# Low-Cost Manufacturing of Monolithic Resonant Piezoelectric Devices for Energy Harvesting Using 3D Printing

**DOI:** 10.3390/nano13162334

**Published:** 2023-08-14

**Authors:** Marcos Duque, Gonzalo Murillo

**Affiliations:** Department of Nano and Microsystems, Instituto de Microelectrónica de Barcelona—Centro Nacional de Microelectrónica (Consejo Superior de Investigaciones Cientificas) (IMB-CNM, CSIC), 08193 Bellaterra, Spain; marcos.duque@imb-cnm.csic.es

**Keywords:** energy harvesting, IoT, 3D printing, piezoelectric, monolithic, magnetic field

## Abstract

The rapid increase of the Internet of Things (IoT) has led to significant growth in the development of low-power sensors. However; the biggest challenge in the expansion of the IoT is the energy dependency of the sensors. A promising solution that provides power autonomy to the IoT sensor nodes is energy harvesting (EH) from ambient sources and its conversion into electricity. Through 3D printing, it is possible to create monolithic harvesters. This reduces costs as it eliminates the need for subsequent assembly tools. Thanks to computer-aided design (CAD), the harvester can be specifically adapted to the environmental conditions of the application. In this work, a piezoelectric resonant energy harvester has been designed, fabricated, and electrically characterized. Physical characterization of the piezoelectric material and the final resonator was also performed. In addition, a study and optimization of the device was carried out using finite element modeling. In terms of electrical characterization, it was determined that the device can achieve a maximum output power of 1.46 mW when operated with an optimal load impedance of 4 MΩ and subjected to an acceleration of 1 G. Finally, a proof-of-concept device was designed and fabricated with the goal of measuring the current passing through a wire.

## 1. Introduction

The large increase in the Internet of Things (IoT) has led to a huge growth in the development of very low-power sensors that communicate with each other and form a large wireless network [1,2,3,4]. The greatest difficulty in the expansion of the IoT is the energy dependence of the sensors, limited by the use of their batteries. For this reason, we are facing a great barrier due to the high cost of replacing batteries and the environmental impact that it entails. A promising solution that provides autonomic energy to the IoT sensor nodes is the harvesting of residual environmental energy, such as movement, vibrations, magnetic fields, light or heat, and converting them into electrical energy (Energy Harvesting (EH)) [1,3,5,6,7,8,9]. Mechanical energy can be converted into electrical energy using piezoelectric transducers [10,11,12,13,14].

In vibrational energy harvesting, piezoelectric materials are one of the most widely used transducers. For the last 20 years, piezoceramics, such as lead zirconate titanate (PZT), have been the dominating materials. After the classification of lead as a toxic material, restrictions on its use have increased, opening the opportunity to investigate new lead-free materials such as polyvinylidene fluoride (PVDF) and its copolymers.

Three-dimensional printing or additive manufacturing (AM) is a process of making three-dimensional solid objects from a digital file [15,16,17,18]. It is increasingly used for the production of any type of custom design in the fields of agriculture, healthcare, automotive and aerospace.

AM technologies are based on three types: Synthesis, in which the temperature of the material is raised without liquefying it to create high-resolution prototypes, fusion, in which powders are fused by means of electron beams, and stereolithography, which uses a method called photopolymerization, which uses an associated ultraviolet laser [19].

According to the American Society for Testing and Materials (ASTM), AM has been divided into seven processes that include:

Blinder jetting (BJ), an additive manufacturing process in which a liquid bonding agent is selectively deposited to join powder materials; directed energy deposition, an additive manufacturing process in which focused thermal energy is used to fuse materials by melting them as they are being deposited; material extrusion, an additive manufacturing process in which material is selectively dispensed through a nozzle or orifice; material jetting, an additive manufacturing process in which droplets of build material are selectively deposited; powder bed fusion (PBF), additive manufacturing process in which thermal energy selectively fuses regions of a powder bed; sheet lamination, an additive manufacturing process in which sheets of material are bonded to form an object; vat photopolymerization, an additive manufacturing process in which liquid photopolymer in a vat is selectively cured by light-activated polymerization.

Fused deposition modeling (FDM) is a material extrusion process used to make thermoplastic parts through heated extrusion and deposition of materials layer by layer. FDM is the most popular technology for 3D printing.

There are many 3D printing materials such as; PLA (polylactic acid), ABS (acrylonitrile butadiene styrene), PVA (polyvinyl alcohol), PP (polypropylene), PLA Tough, CPE (co-polyester), PET-G (glycol-modified polyethylene terephthalate), Nylon (polyamide) and PC (polycarbonate), but the most common material is PLA. PLA is a biodegradable and recyclable polymer that is a reliable and easy-to-print material that can be printed at low temperatures. It has a low shrinkage factor and does not require the use of a heated build plate. A special type of PLA, called Tough PLA, combines the ease of printing with increased mechanical performance. Specifically, tough PLA has increased impact resistance, avoiding the typical brittle failure of normal PLA. Tough PLA is used for functional prototyping tooling and manufacturing aids.

Through 3D printing, it is possible to create monolithic functional structures (i.e., resonators, sensors, energy harvesters, etc.). This reduces costs as it eliminates the need for subsequent assembly tools. It is a technology that does not require additional molds, allowing for quick design variations and the creation of more complex structures. Furthermore, thanks to computer-aided design (CAD), the devices can be specifically tailored to the environmental conditions of the application. Figure 1 illustrates the advantage of integrating piezoelectric materials by 3D printing to create monolithic 3D-printed piezoelectric devices. In this way, it is possible to adapt the harvester to our needs, making it fully customizable, and the cost of the final device is reduced because subsequent assembly is not required.

In the literature, there are several studies on the use of 3D printing to create energy harvesters. In most of these works, the printed piece is used as a support and the harvester is assembled later [20,21,22,23,24,25,26,27,28,29,30]. In this work, a resonant piezoelectric energy harvester has been designed and manufactured, using 3D printing by hybridizing this technology with a piezoelectric material (PVDF) to achieve a monolithic energy harvester. In addition, a study and optimization of the device have been carried out by means of finite element modeling (COMSOL Multiphysics^®^). Finally, a use-case resonator was designed and fabricated with the goal of measuring the current carried by a wire.

## 2. Material and Methods

### 2.1. Structural and Functional Materials

The piezoelectric material used in the harvester is a commercially available PVDF film with two parallel electrodes, from TE Connectivity. According to the manufacturer’s specification, the PVDF material has a thickness of 110 µm and 6 µm for each silver ink electrode. The piezoelectric coefficient of the commercial PVDF is −33 × 10^−12^ m/V, according to the datasheet. The selected material for the structural harvester is Ultimaker Tough PLA green from Ultimaker, with a Young’s modulus of 2.8 ± 0.15 GPa and a density of 1.22 g/cm^3^. A neodymium (Nd) mass with a density of 7 g/cm^3^, from Supermagnete, was used.

### 2.2. Finite Element Modelling

Finite element modeling (FEM) of the energy harvester was conducted using COMSOL Multiphysics^®^ 6.0. The AC/DC (Electrostatics, Electrical Circuits and Piezoelectricity) and Structural Mechanics (Solid Mechanics and Piezoelectricity) modules were utilized for the simulation. The model was created using a fine triangular mesh, consisting of over 25,000 elements.

### 2.3. Manufacturing Process and Characterizations

The Ultimaker S3 printer with AA 0.25 mm nozzle was employed for the manufacturing process, allowing printing layers from 60 µm to 150 µm thick.

The initial characterization involved a stepper motor from Zaber Technologies LSQ075B-T3-MC03 (Vancouver, BC, Canada), capable of performing movements from 1 mm to 6 mm on the *Z*-axis. Concurrently, the voltage generated by the harvester was measured using a Sourcemeter Keithley 2470 (Cleveland, OH, USA). A force sensor Mark-10 (Copiage, NY, USA) MR03-20 fixed onto the stepper motor recorded the data with the assistance of a dynamometer Mark-10 M5I. An ad-hoc LabVIEW program controlled the setup comprising the devices.

The second electrical characterization was conducted using an electrodynamic shaker VR9500 from Vibration Testing Controller (Jenison, MI, USA)to emulate environmental vibrations at different input acceleration magnitudes. This process was controlled by a customized MATLAB program, enabling automation, acquisition, and processing of voltage measurements. All data were measured using an acquisition system from National Instruments (Austin, TX, USA) PCI-6132 and a BNC-2110 terminal block.

The piezoelectric coefficient of the material was measured using a Wide-Range d33 piezometer from APC International, Ltd. (Mackeyville, PA, USA).

## 3. Results

### 3.1. Design of the Harvester

#### 3.1.1. Definition Fixed Parameters

There are different sources for energy harvesting, such as vibrations, human motion, residual magnetic field, etc. In this case, we have chosen the magnetic fields generated by a current-carrying conductor. For this reason, the resonance frequency is fixed at 50 Hz (European electric grid frequency).

In order to study structure optimization, an effective area and resonance frequency of the device must be set. In this case, the dimensions are 50 mm × 40 mm (length × width).

#### 3.1.2. Device Structure

There are different types of structures for vibrational harvesters, such as cantilevers, doubly supported beams or diaphragms. The cantilever results in the highest power output when used as a piezoelectric energy harvester. Furthermore, the cantilever structure leads to the lowest resonance frequency for a given size. For this reason, the cantilever structure has been chosen.

The resonance frequency of a spring-mass system can be expressed as [31,32,33]:(1)$$f_{r}=\frac{1}{2\pi }\sqrt{\frac{k_{\mathrm{eff}}}{m_{\mathrm{eff}}}},$$

Here, k_eff_ and m_eff_ represent the effective suspension stiffness and the effective mass values, respectively.

The neutral axis of a beam is the layer in the cross-section of a beam that experiences no longitudinal strain under bending. For composite beam structures, the effect of the different materials of each layer must be summed up to find the neutral axis:(2)$$Z=\frac{\sum_{i=1}^{n}z_{i}Y_{i}h_{i}}{\sum_{i=1}^{n}Y_{i}h_{i}},$$
where Z is the neutral axis position of the composite beam, Z_i_ is the height of the centroid of layer i, h_i_ is the thickness of layer i and Y_i_ is Young’s modulus of layer i.

### 3.2. Shape Optimizations and Finite Elements Modeling

#### 3.2.1. Obtaining Resonance Frequency

In addition to the reduction in the manufacturing cost, by having a monolithic device, anchor losses are avoided due to the improved resonator anchor structure. To achieve the desired resonance frequency, a mass with a very large thickness was required due to the low density of the PLA. Furthermore, to achieve actuation with magnetic fields by means of Lorentz’s force, a ferromagnetic material is needed as a cantilever tip mass. For this reason, Nd magnets have been used as an inertial mass.

As shown in Figure 2a, our harvester consists of a PLA cantilever with a length composed of the length of the beam (L_b_), the mass (L_m_) and the length of the cantilever (L_c_). Moreover, the thickness of the beam has been defined as (T_b_), the thickness of the mass as (T_m_) and the depth of the piezoelectric material with respect to the surface as (P_d_). 

The 3D model shown in Figure 2b consists of the energy harvester with the piezoelectric layer inside the structure and an external circuit with a load resistor to calculate the power. 

A piezoelectric material produces an electrical charge from mechanical stress. The d33 coefficient represents the piezoelectric charge coefficient along a specific direction, reflecting the material’s response to applied stress. It can be expressed as:(3)$$d33=\frac{\Delta Q}{\Delta S},$$
where ΔQ is the induced charge density and ΔS is the applied stress along a particular axis.

From Equation (2), it can be deduced that the highest stress resulting from a beam deformation in a cantilever is located on the surface. In Figure 2c, the stress distribution is shown when the device resonates at its resonance frequency. It can be observed that the stress on the surface of the material is at its maximum, while it is zero at the center of the beam. Consequently, during the manufacturing process, it is crucial to position the encapsulated piezoelectric material as close to the surface as feasible. However, there is a trade-off, if the piezoelectric layer is too close to the surface, it will not be properly embedded and can get delaminated. 

A parametric simulation of L_b_ and L_m_ is performed, to find the different combinations to obtain a resonance frequency of 50 Hz. The dimensions of the magnasets are more restrictive because they are defined by the manufacturer. A parametric study from 10 mm to 40 mm of L_m_ is performed. Table 1 shows the different dimension combinations obtained to achieve a resonance frequency of 50 Hz.

#### 3.2.2. Study of Generated Power

The voltage and power of the different dimensions are studied using an external circuit that includes a resistor. Performing a sweep of the external resistor, the optimal load resistor can be determined to achieve maximum power. As shown in Figure 3a, the lower ratio between L_b_ and L_c,_ the higher the power output. Furthermore, a smaller piezoelectric area is needed, leading to a decrease in device cost since PVDF is the most expensive material in the harvester.

A parametric thickness of the beam under the piezoelectric sheet was simulated to improve the power generated by the piezoelectric generator. Due to the increase in L_b_, the natural frequency of the device increases. To compensate for this augment, the density or T_m_ must be raised. In this occasion, the density of the mass has been augmented to compensate for the frequency. In this way, the dimension of the mass is always the same and simplifies the meshing process during the simulation. As can be seen in Figure 3b, the greater T_b_, the greater the power generated.

In the next Table 2, different T_m_ can be seen to compensate for the increased T_b_. In addition, the data of the maximum generated power, volume and power density of each harvester obtained from the simulation are also shown.

With the data obtained in the simulation shown in Table 2, the volume and cost of each harvester are calculated. As seen in Figure 4a, the cost and the volume of the harvester are closely related because most of the volume is mass. Furthermore, it can be seen how the highest power density is for a T_b_ of 1.5 mm.

Finally, a parametric simulation of the depth of the piezoelectric material regarding the surface has been performed. As can be seen in Figure 4b and as mentioned above in the design section. The closer the material is to the surface, the more power it generates since it is farther from the neutral line of the beam.

With the results obtained from the simulations, it can be concluded that the best specification for the harvester is the design with a dimension of 10.2 mm × 40 mm (L_c_ and L_m_). In addition, to obtain a higher power density, the beam should be 1.5 mm thick and the piezoelectric material should be as close to the surface as possible. 

### 3.3. Manufacturing Process

In 3D printing, as can be seen in Figure 5 the process begins with designing the desired piece using 3D design software (Tinkercard, https://www.tinkercad.com), typically CAD software. Once the design is completed, the next step is exporting the file in STL (Stereolithography) format. The STL file contains all the necessary information about the 3D object. Subsequently, the CAD model is sliced into layers using slicing software Ultimaker Cura 4.13.1. Finally, the object is printed by transferring the file, positioning and printing the design layer-by-layer.

Our manufacturing process is very similar to a typical 3D object; however, two additional steps are required to introduce the bottom electrode and the PVDF material. The first step, shown in Figure 6a, is the printing of the brim around the object to hold the edges of your piece and subsequently printing the layer that encapsulates the piezoelectric material within the structure. As previously mentioned in the generated power simulations, a thinner layer leads to greater power output. Since the minimum layer the nozzle can print is 60 µm, and to ensure good quality and avoid pushing the limits of the printer head, it was decided to print two or three layers of 100 µm. It is crucial to maintain high quality in this layer because it contributes to the stress received by the piezoelectric material. Additionally, a hole is left on the material’s surface to place the lower copper electrode.

Subsequently, a copper piece is inserted to establish the connection with the bottom electrode, and the printing process continues depositing the piezoelectric material layer. At this point, the printing process is paused again to add the piezoelectric material, followed by the printing of the remaining structure. During this final stage of the printing, the rest of the beam and support with two holes to secure the harvester are created. Upon completing the print, gold connectors are inserted to facilitate the connection between the two electrodes of the piezoelectric material. In Figure 7, it can be seen the images of the steps of our 3D printing process to create the harvester. Three harvesters were manufactured with the piezoelectric material at 200 µm from the surface and other three devices with 300 µm.

### 3.4. Characterization

#### 3.4.1. Electrical Characterization

Once the harvesters are manufactured, we proceed to the electrical characterization of our final devices. Two electrical characterizations have been conducted. The first characterization, shown in Figure 8a, involves a stepper motor that performs movements from 1 mm to 6 mm of distance on the Z-axis. Simultaneously, the voltage generated by the harvester is measured using a Sourcemeter. A force sensor, attached to the stepper motor, records these data through a dynamometer. An ad-hoc LabVIEW program controls all the setup comprising these devices. As can be seen in Figure 8b, a displacement of 6 mm along the Z-axis results in an average voltage of 71.7 V for the harvester with the piezoelectric material positioned 300 µm far from the surface. Reducing the distance of the piezoelectric material from 300 µm to 200 µm increases the average voltage to 85.2 V, representing a 19% increment in the maximum voltage generated. According to the force data represented in Figure 8c, the depth of the piezoelectric material does not affect the force needed to achieve 6 mm of travel at the tip of the cantilever.

For the second electrical characterization, as can be shown in Figure 9a, a magnetic mass is attached to adjust the resonance frequency to 50 Hz. An electrodynamic shaker is used to emulate environmental vibrations at different input acceleration levels. This is controlled through an ad-hoc MATLAB program that allows automatizing, acquiring and processing the voltage measurements. All these data are captured by an acquisition system. Subsequently, fixing a certain acceleration magnitude, a sweep of load resistors values is performed. This allows us to determine the optimal load resistor value for the highest generated power.

Due to the voltage limitations of the acquisition system, which cannot measure voltages exceeding 10 V, the load resistance sweep is initially carried out with a small acceleration of 0.1 G, ensuring that it remains below 10 V. Figure 9b shows that the optimal load resistance for maximum power generation is 4 MΩ, resulting a maximum power of 4.4 µW.

Once the optimal load resistance is known, use a voltage divider, composed of a set of 500 KΩ and 3.5 MΩ resistors. It is possible to measure the voltages generated by harvesters with accelerations greater than 0.1 G.

The voltage generated by the piezoelectric generator was measured while performing an acceleration sweep ranging from 0.1 G to 1 G, as shown in Figure 9c. For an acceleration of 1 G, the harvester with the piezoelectric material positioned 300 µm from the surface achieved an average voltage of 62.3 V. By reducing the distance of the piezoelectric material to 200 µm, an average voltage of 75.6 V is achieved, resulting in a 21.4% increase in the maximum voltage generated. Using the aforementioned voltage data, the maximum power for the different accelerations was calculated and plotted in Figure 9d. It can be observed that reducing the difference between the two depths of the piezoelectric material resulted in an increase of up to 47.3% in the maximum power generated. This led to a maximum output power of 1.46 mW.

#### 3.4.2. Physical Characterization of PVDF Material

An inspection using a scanning electron microscope (SEM) was performed to validate the thickness of materials provided by the manufacturer and, as can be seen in Figure 10a, the measured thickness closely matches the specifications.

Using a piezometer, we can measure the exact coefficient of our PVDF piece. As shown in Figure 10b, the coefficient is the same as the manufacturer’s specifications.

#### 3.4.3. Material Measurement of Young’s Modulus of Our Harvester

The Young’s modulus of the used PLA is 2.8 ± 0.15 GPa, as stated in the technical datasheet. When introducing a material within the PLA, the overall effective Young’s modulus can be changed. Therefore, the Young’s modulus of the harvesters will be calculated based on the force data measured with a stepper motor displacement from the previous section.

A displacement of a cantilever beam δ is related to the applied load, P and the Young modulus, E by the next Equation (4):(4)$$\delta =\frac{1}{3}\;\frac{{\mathrm{PL}}^{3}}{\mathrm{EI}},$$
where L is the length of the cantilever, and I the second moment of area (moment of inertia).

For a prismatic beam with a rectangular section (depth h and with w) the value of the second moment inertia of cantilever beam I is given by:(5)$$I=\frac{{\mathrm{wh}}^{3}}{12}=\frac{0.04\;m\times 0.0015\;m^{3}}{12}=11.25\times 10^{-12}\;m^{4},$$

From Equation (4) and using the force data measured with stepper motor displacement, the Young’s modulus of harvesters is calculated:(6)$$E=\frac{1}{3}\;\frac{{\mathrm{PL}}^{3}}{\delta I}=\frac{4.5\;N\times 0.05\;m^{3}}{3\times 0.006\;m\times 11.25\times 10^{-12}\;m^{4}}=2.78\;\mathrm{GPa},$$

From Young’s modulus obtained from the above Equation (6), it can be seen that the modulus is within the manufacturer’s specifications and that the piezoelectric material has little effect on the final Young’s modulus of the harvesters.

### 3.5. Finite Elements Simulation 

The energy harvesters were simulated again with COMSOL Multiphysics^®^ to compare the results of the electrical characterization with a theoretical model. Physical parameters such as Young’s modulus, thickness, coefficient of the piezoelectric material, and harvester dimensions were included in the simulation to achieve a realistic representation.

A parametric sweep ranging from 48 Hz to 54 Hz was performed to determine the resonance frequency of the resonators with an optimal load resistance of 4 MΩ. The acceleration used for this simulation was 0.5 G. It can be seen, in Figure 11a, the resonance frequency for the COMSOL Multiphysics^®^ harvesters was approximately 50.5 Hz, while for the electrical characterization of the manufactured harvesters yielded a resonance frequency of 49.5 Hz. Initially, the harvesters were tuned to resonate at 50 Hz with an acceleration of 0.1 G. However, due to spring softening with increasing acceleration, the resonance frequency is shifted from 0.5 Hz to 49.5 Hz.

As in the electrical characterization with the shaker, a sweep of the load resistance is performed to calculate the maximum power. As shown in the Figure 11b, the maximum power generation is around 5 MΩ, with a maximum power of 283 µW for the harvester with a 300 µm distance of the piezoelectric material and 464 µW for the 200 µm distance. These values are very similar to the maximum powers measured in Figure 9c, with 288 µW and 462 µW, respectively.

### 3.6. Example of Application

As mentioned earlier, the energy harvester can be customized using CAD software (e.g (Tinkercard, https://www.tinkercad.com) to adapt its specifications for various application scenarios. An example application for our device would be its use as a sensor rather than an energy harvester. In this particular case, it can be employed to measure the current flowing through a wire in a high-voltage tower. When an electric current passes through the wires, it generates a magnetic field that is directly proportional to the current flowing through the wire. By using the magnetic mass, our device can be made to resonate, and depending on the current it passes, we can measure an instantaneous proportional voltage in our sensor.

In this example application, the maximum dimensions for the sensor are 30 mm × 20 mm with a height of 10 mm. The resonance frequency for this resonator is the same as the frequency of the current flowing through the wire being measured, which is 50 Hz. The magnets for the magnetic mass have dimensions of 10 mm × 10 mm and a thickness of 2.7 mm. The support will be fixed at dimensions of 20 mm × 10 mm and a height of 10 mm to ensure good anchoring and easy fastening. A simulation is performed to obtain the resonance frequency at 50 Hz. As shown in Figure 12a, measurements of 12 mm × 20 mm and a thickness of 700 µm for the cantilever beam are obtained. Two magnets will be required for the magnetic mass. Figure 12b shows the stress distribution in the beam when the cantilever is resonating at an acceleration of 0.1 G.

As can be shown in Figure 13a,b, the simulated sensor is mounted inside a sensor node provided by the startup company Energiot Devices SL (www.energiot.com). To test the sensor in the laboratory, two heaters with two power selections have been connected. This setup allows for the generation of four different currents depending on the selected power: 1500 W (6.5 A), 3000 W (13 A), 4500 W (19.5 A) and 6000 W (26 A). As can be seen in Figure 13c, the different current cases passing through the cable can be measured using the corresponding voltage readings, demonstrating a linear response.

Furthermore, the magnetic field can also be used to power wireless sensors that measure various parameters such as magnetic field, ambient and line temperature, acceleration, tilt, humidity and light intensity and transmit the gathered information to the cloud.

## 4. Conclusions

In this article, we designed, manufactured and electrically characterized a resonant piezoelectric energy harvester by exploiting 3D printing technology and encapsulating PVDF within the structure, resulting in a monolithic harvester. With this, an economically viable energy harvester was achieved that mechanically improves the anchoring of the resonator thanks to its structure attached to the fixing support. The device was optimized through the finite element simulation software COMSOL Multiphysics^®^ 6.0.

Furthermore, we demonstrated an example of the practical application of our resonator by designing and fabricating a specific use case for measuring the current passing through a wire using magnetic fields. Through 3D printing, not only did it enable the manufacturing of a compact and monolithic harvester but also facilitated the customization and optimization of the device for specific requirements.

Overall, this work contributes to the advancement of piezoelectric energy harvesting technology by providing a practical approach to designing and manufacturing resonant harvesters using 3D printing. The demonstrated use case application shows the potential to use such devices in a myriad of real-world scenarios.

Finally, this article represents a small contribution to the ongoing trends and future sustainability of self-powered sensors. By harvesting ambient energy, self-powered sensors have the potential to reduce the environmental impact associated with batteries. Furthermore, current advancements are paving the way for more efficient and reliable self-powered sensors. However, further research and development are needed to overcome challenges such as energy conversion efficiency and long-term reliability improvement.

## Figures and Tables

**Figure 1 nanomaterials-13-02334-f001:**
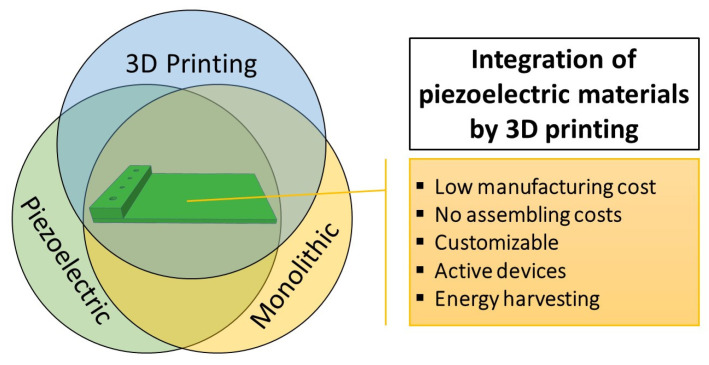
Advantages of using a hybrid solution of 3D printing and energy harvesting.

**Figure 2 nanomaterials-13-02334-f002:**
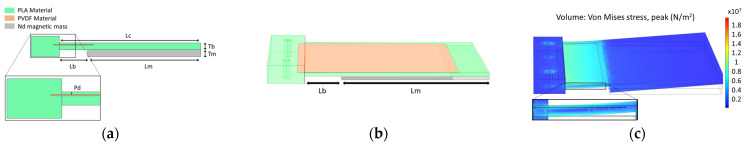
(**a**) Schematic of the cross-section of the energy harvester with the dimensions; (**b**) 3D design used and (**c**) stress distribution when the harvester resonates at its resonance frequency and detail of the cross-section showing the neutral axis in COMSOL Multiphysics^®^.

**Figure 3 nanomaterials-13-02334-f003:**
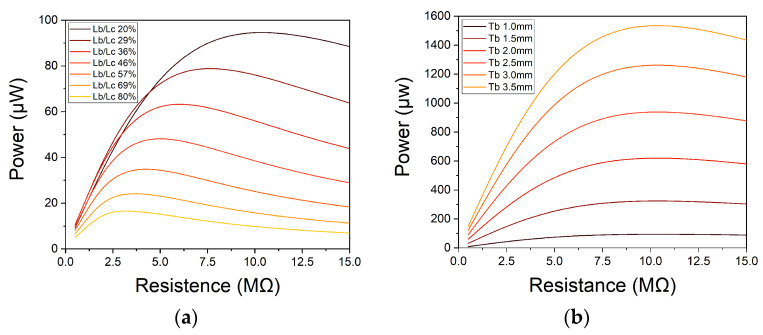
(**a**) Maximum power generated by the device with different L_c_/L_b_ ratios; (**b**) Maximum power generated modifying the neutral axis of the beam increasing T_b_ and the density of the mass to maintain the same resonance frequency.

**Figure 4 nanomaterials-13-02334-f004:**
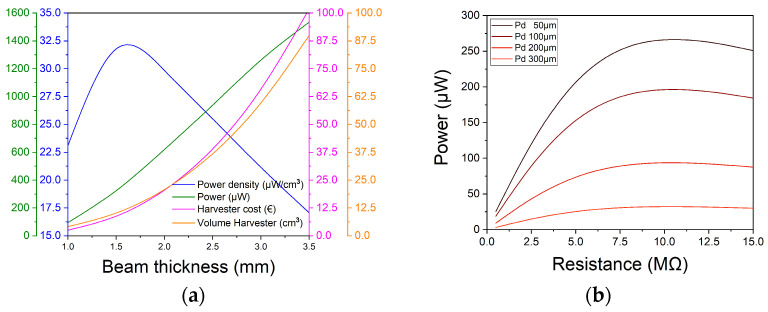
(**a**) Graphic of the maximum power generated, power density, harvester cost and volume of the harvester as a function of beam thickness; (**b**) Power curve generated by the harvester as a function of the depth of the piezoelectric material with respect to the surface.

**Figure 5 nanomaterials-13-02334-f005:**
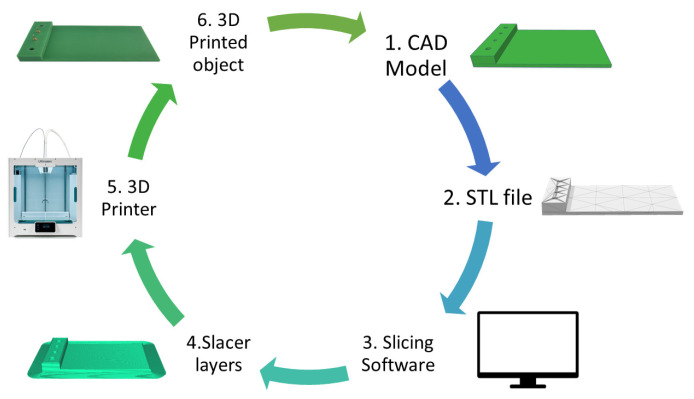
Basic process of 3D printers to create 3D objects.

**Figure 6 nanomaterials-13-02334-f006:**
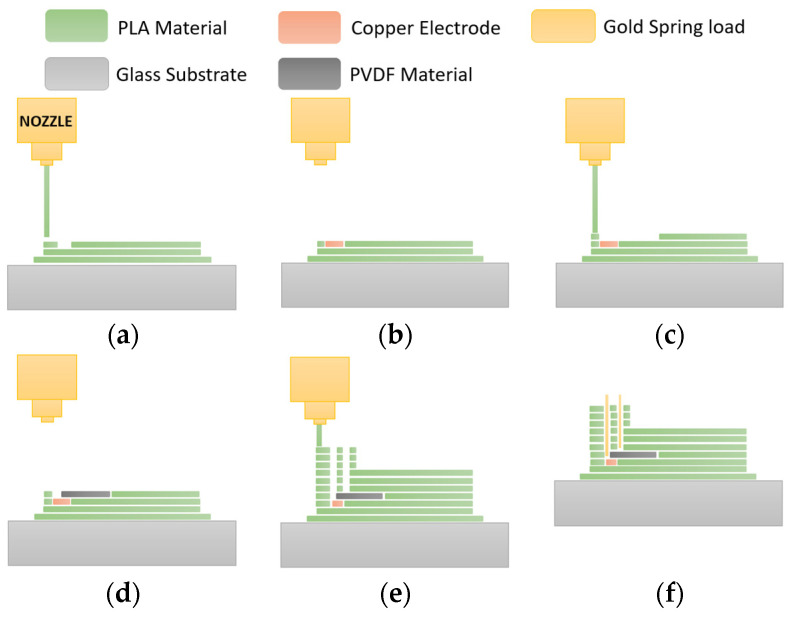
(**a**) Schematic of our 3D printing process and the sequences of steps to create the device (**a**) PLA base layer printing for encapsulation of the piezoelectric material; (**b**) placement of copper material to make the connection of the bottom electrode; (**c**) PLA layer printing for later assembly of the piezoelectric material; (**d**) placement of the piezoelectric material (PVDF); (**e**) printing of the rest of the PLA structure leaving two holes for subsequent connections and (**f**) insertion of gold connectors for the connection of the two electrodes of the piezoelectric material.

**Figure 7 nanomaterials-13-02334-f007:**
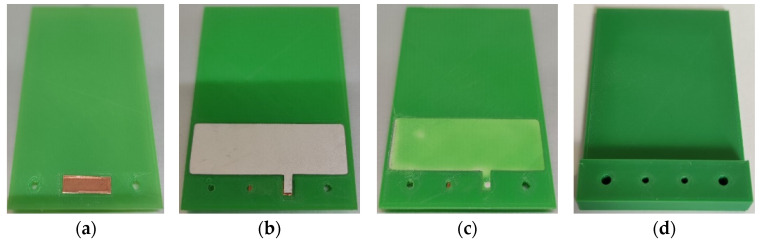
Images of steps of our 3D printing process to create the harvester (**a**) PLA base layer printing for encapsulation of the piezoelectric material and placement of copper material to make the connection of the bottom electrode; (**b**) placement of the piezoelectric material; (**c**) next layer printed after the placement of the PVDF to see how it is encapsulated and (**d**) printing of the rest of the PLA structure leaving two holes for subsequent connections.

**Figure 8 nanomaterials-13-02334-f008:**
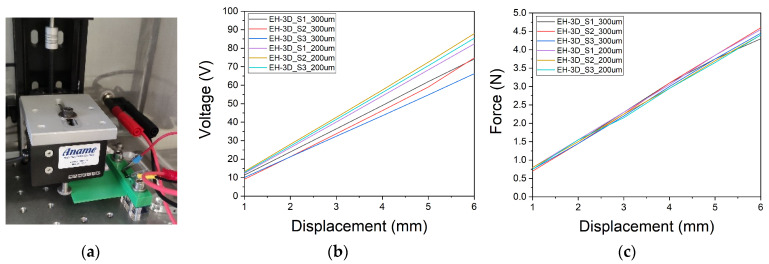
(**a**) Setup for electrical characterization using a stepper motor, sourcemeter and dynamometer; (**b**) Maximum voltage achieved with a displacement of the stepper motor from 1 mm to 6 mm of distance on the Z-axis and (**c**) Force measured with a displacement of the stepper motor from 1 mm to 6 mm of distance on the Z-axis.

**Figure 9 nanomaterials-13-02334-f009:**
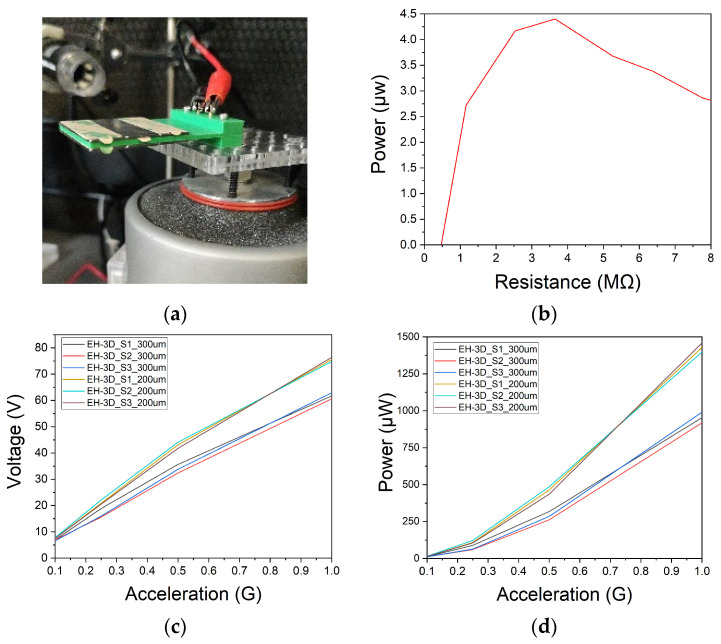
(**a**) Setup for electrical characterization using an electrodynamic shaker; (**b**) Optimal load resistance for maximum power generation with 0.1 G of acceleration; (**c**,**d**) Voltage and maximum power for an acceleration sweep from 0.1 G to 1 G and using a frequency of 50 Hz and an optimal load resistance of 4 MΩ.

**Figure 10 nanomaterials-13-02334-f010:**
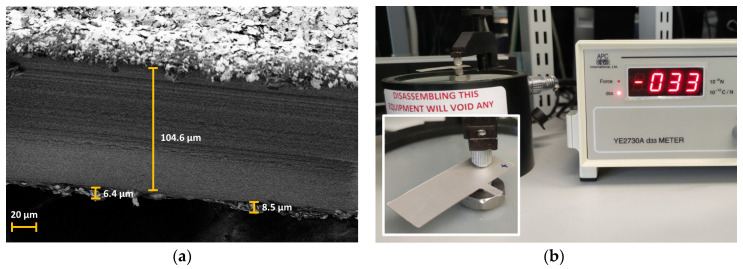
(**a**) SEM image of the thicknesses of the layers of the piezoelectric material; (**b**) Image of the measurement of the piezoelectric coefficient.

**Figure 11 nanomaterials-13-02334-f011:**
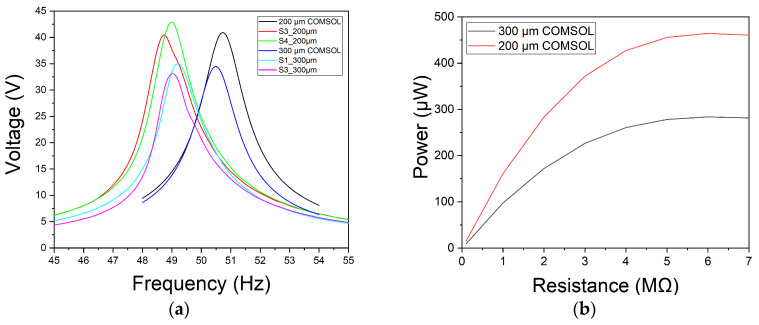
Simulated results of the piezoelectric device by using COMSOL Multiphysics^®^: (**a**) Comparative of simulation and electrical characterization of the maximum open-circuit voltage that can be generated by the piezoelectric generator submitted to a frequency sweep with an acceleration of 0.5 G; (**b**) A comparison of the maximum power generated in relation to the depth of the piezoelectric material and the load impedance.

**Figure 12 nanomaterials-13-02334-f012:**
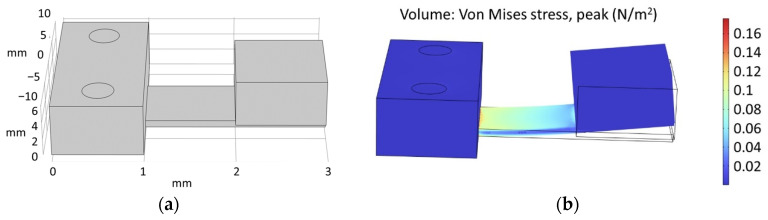
Simulated results of the piezoelectric current sensor by using COMSOL Multiphysics^®^: (**a**) Structure and dimensions used in the simulation; (**b**) 3D FEM simulation of the stress distribution in the cantilever beam.

**Figure 13 nanomaterials-13-02334-f013:**
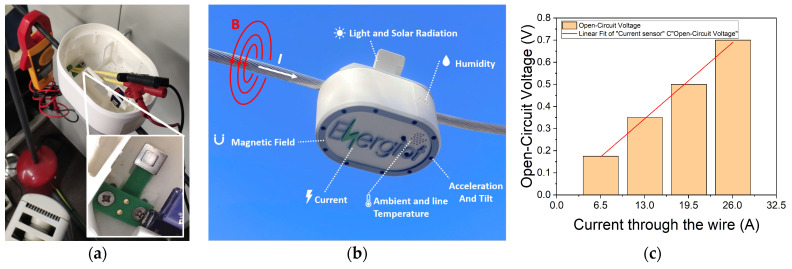
(**a**,**b**) Piezoelectric current sensor manufactured using 3D printing technology mounted inside a sensor node from the start-up Energiot; (**c**) Open circuit voltage generated by the piezoelectric sensor, depending on the selected power of two heaters, which can obtain up to four different currents (1500 W, 3000 W, 4500 W and 6000 W).

**Table 1 nanomaterials-13-02334-t001:** Comparative with the different combinations of the parameters to obtain the fixed resonance frequency at 50 Hz.

L_b_ (mm)	L_m_ (mm)	L_c_ (mm)	Ratio L_b_/L_c_ (%)	Frequency (Hz)
41.20	10.00	51.20	80	50.04
33.10	15.00	48.10	69	49.95
26.80	20.00	46.80	57	50.02
21.60	25.00	46.60	46	50.00
17.15	30.00	47.15	36	50.01
13.35	35.00	45.35	29	50.05
10.20	40.00	50.20	20	49.99

**Table 2 nanomaterials-13-02334-t002:** Comparative with the different thicknesses of mass to compensate for the increased thickness of the beam and data of the maximum power generated, volume and power density of each harvester.

T_b_ (mm)	T_m_ (mm)	Power (µW)	Volume (cm^3^)	Power Density (µW/cm^3^)
1.00	1.00	95	4.10	23.08
1.50	3.48	324	10.20	31.77
2.00	8.13	620	20.80	29.85
2.50	15.46	938	36.80	25.50
3.00	26.12	1261	59.60	21.15
3.50	40.40	1534	89.90	17.06

## Data Availability

Data are contained within the article.

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
