# Peer review of "Low-Cost Manufacturing of Monolithic Resonant Piezoelectric Devices for Energy Harvesting Using 3D Printing"

_nanomaterials, 2023, doi:10.3390/nano13162334_

Round 1
Reviewer 1 Report
There are many papers and presentations on the improvement of harvester power generation through frequency matching, and research has been conducted for many years.
If the contents are too insufficient to be called a review paper, it is questionable what kind of relevance the contents of Figure 12 are in particular.
The whole thing needs fixing.
Author Response
There are many papers and presentations on the improvement of harvester power generation through frequency matching, and research has been conducted for many years.
If the contents are too insufficient to be called a review paper, it is questionable what kind of relevance the contents of Figure 12 are in particular.
We appreciate the feedback from the reviewer. However, we would like to highlight that this paper is not focused on the improvement of the power generation but the use of 3D printing technology to fabricate monolithic piezoelectric devices (e.g. energy harvesters, sensors, etc.).
The reviewer pointed out that this paper cannot be considered a review paper, however this is not indeed a review paper but a research article. The figure 12 has been improved to highlight the final application of one prototype fabricated with the proposed technology.
Reviewer 2 Report
This article describes piezoelectric resonant energy harvester has been designed, fabricated, and electrically characterized, I recommend major revisions for the manuscript prior to publication in Nanomaterials.
Major comments:
1. There are a number of articles about various type of energy harvester for self-powered applications using sensors and IOT applications. In short, the manuscript failed to bring new insights to this field.
2. The part of the discussion is not comprehensive, and the related literature is not mentioned, and some experimental data have not been updated in time from the latest literature.
3. In current work, I have not seen the author's trends and future sustainability of self-powered sensors. More focus and content are on the work that has been published, and no substantive comments have been given. Therefore, I cannot get the direction or challenge of the future development of the self-powered sensors. The authors repeated some points many times in their manuscript, resulting in poor readability.
4. Why 3D printing and hybridizing this technology with other rapid prototyping techniques for fabrication of piezoelectric resonant energy harvester.
5. The working phenomenon of this kind piezoelectric resonant energy harvester can be discussed with the help of the COMSOL simulation. The conversion efficiency can be evaluated.
6. How about the life-time of this kind of piezoelectric resonant energy harvester?
7. The readability of the paper still needs to be further improved for the language quality and
organization.
In the introduction part of the manuscript is too general. Authors should include relevant literatures in the introduction section to emphasize the significance of their work. Some of the important literature needed to compare are listed below before acceptance of manuscript.
References:
[1] Sci Rep 13, 5283 (2023). https://doi.org/10.1038/s41598-022-23574-2
[2] Nano Energy 53 (2018) 1003–10191004, https://doi.org/10.1016/j.nanoen.2018.09.032
[3] Journal of Materials Science volume 57, pages4399–4440 (2022)
[4] Sensors 2020, 20(10), 2925; https://doi.org/10.3390/s20102925
This article describes piezoelectric resonant energy harvester has been designed, fabricated, and electrically characterized, I recommend major revisions for the manuscript prior to publication in Nanomaterials.
Major comments:
1. There are a number of articles about various type of energy harvester for self-powered applications using sensors and IOT applications. In short, the manuscript failed to bring new insights to this field.
2. The part of the discussion is not comprehensive, and the related literature is not mentioned, and some experimental data have not been updated in time from the latest literature.
3. In current work, I have not seen the author's trends and future sustainability of self-powered sensors. More focus and content are on the work that has been published, and no substantive comments have been given. Therefore, I cannot get the direction or challenge of the future development of the self-powered sensors. The authors repeated some points many times in their manuscript, resulting in poor readability.
4. Why 3D printing and hybridizing this technology with other rapid prototyping techniques for fabrication of piezoelectric resonant energy harvester.
5. The working phenomenon of this kind piezoelectric resonant energy harvester can be discussed with the help of the COMSOL simulation. The conversion efficiency can be evaluated.
6. How about the life-time of this kind of piezoelectric resonant energy harvester?
7. The readability of the paper still needs to be further improved for the language quality and
organization.
In the introduction part of the manuscript is too general. Authors should include relevant literatures in the introduction section to emphasize the significance of their work. Some of the important literature needed to compare are listed below before acceptance of manuscript.
References:
[1] Sci Rep 13, 5283 (2023). https://doi.org/10.1038/s41598-022-23574-2
[2] Nano Energy 53 (2018) 1003–10191004, https://doi.org/10.1016/j.nanoen.2018.09.032
[3] Journal of Materials Science volume 57, pages4399–4440 (2022)
[4] Sensors 2020, 20(10), 2925; https://doi.org/10.3390/s20102925
Author Response
- There are a number of articles about various type of energy harvester for self-powered applications using sensors and IOT applications. In short, the manuscript failed to bring new insights to this field.
This manuscript is focused on the use of 3D printing technology to fabricate monolithic piezoelectric devices (e.g. energy harvesters, sensors, etc.). The novelty of this approach is how to use this rapid prototyping technique to fabricate in a single component a resonant structure together with a piezoelectric layer with two electrodes. We have proposed a change in the name to make sure this point. Also, we have modified the abstract and introduction to highlight the novelty of this work.
- The part of the discussion is not comprehensive, and the related literature is not mentioned, and some experimental data have not been updated in time from the latest literature.
According to the reviewer's suggestions, a significant portion of the introduction has been modified and references to related literature have been added.
- In current work, I have not seen the author's trends and future sustainability of self-powered sensors. More focus and content are on the work that has been published, and no substantive comments have been given. Therefore, I cannot get the direction or challenge of the future development of the self-powered sensors. The authors repeated some points many times in their manuscript, resulting in poor readability.
We thank the reviewer’s comment. Firstly, we have improved the readability of the manuscript by rewriting and rearranging large part of the content (repeated points have been removed). The authors’ point of view about the trends and future sustainability of self-powered sensors have been added with some new comments.
- Why 3D printing and hybridizing this technology with other rapid prototyping techniques for fabrication of piezoelectric resonant energy harvester.
Following the reviewer's suggestions, the introduction has been extended, providing a better explanation for the rationale behind the hybridization of 3D printing and the piezoelectric material. The main reason is that monolithic (single piece) devices can be fabricated with an integrated piezoelectric functionality. In addition, because of the customization capabilities of the 3D printing the electromechanical properties of the devices can be ad-hoc designed, allowing applications to hundreds of use cases and device optimization.
- The working phenomenon of this kind piezoelectric resonant energy harvester can be discussed with the help of the COMSOL simulation. The conversion efficiency can be evaluated.
According to the reviewer's suggestions, Figure 2c has been added, along with an explanation of how a piezoelectric material operates. The stress distribution within the material has been also shown.
- How about the life-time of this kind of piezoelectric resonant energy harvester?
To validate the durability of this harvester, it was continuously operated for over 15 days. It was confirmed that the device can withstand more than 65 million cycles without any degradation in its output power. PLA is not the best material in terms of durability, due to the low temperature resistance (< 60 ºC). However, depending on the final application, there are many other materials with improved mechanical, thermal and chemical properties (ABS, PEEK, Nylon, etc.) that can be used.
- The readability of the paper still needs to be further improved for the language quality and organization.
Following the reviewer's recommendations, the entire document has been revised and restructured to enhance its readability and overall quality.
- In the introduction part of the manuscript is too general. Authors should include relevant literatures in the introduction section to emphasize the significance of their work. Some of the important literature needed to compare are listed below before acceptance of manuscript.
Following the reviewer's recommendations, the introduction has been significantly expanded to substantiate the motivation and innovation of this study. Furthermore, the references provided by the reviewer, including essential and relevant literature, have been incorporated into the article.
Reviewer 3 Report
This manuscript discusses the challenge of the energy dependency of low-power sensors in the Internet of Things (IoT) and proposes energy harvesting from ambient residual energy as a solution. The focus of this work is on the design, fabrication, and characterization of a piezoelectric resonant energy harvester using 3D printing and other rapid prototyping techniques. The physical characterization of the piezoelectric material and the resonator is carried out, and a study and optimization of the device is done using finite element modeling. Finally, an example resonator application is fabricated to measure the current passing through a wire. I recommend this manuscript be published in the Journal of " Nanomaterials" with the following modifications.
1. Specific quantitative results should be given in the abstract for the physical characterization and piezoelectric properties of the device.
2. In the Introduction section the authors need to present the technical problems to be solved in comparison with previous work and the innovation points of this study.
3. The author should combine the information related to the materials and testing instruments in a separate paragraph.
4. The labels in Figures 2a-b and 3b cover part of the curve and need to be modified.
5. The scales of 6.4um and 8.5um in Figure 9a are the same length and need to be checked and corrected.
6. The tick marks for the upper and right axes should be removed.
Minor editing of English language required.
Author Response
This manuscript discusses the challenge of the energy dependency of low-power sensors in the Internet of Things (IoT) and proposes energy harvesting from ambient residual energy as a solution. The focus of this work is on the design, fabrication, and characterization of a piezoelectric resonant energy harvester using 3D printing and other rapid prototyping techniques. The physical characterization of the piezoelectric material and the resonator is carried out, and a study and optimization of the device is done using finite element modeling. Finally, an example resonator application is fabricated to measure the current passing through a wire. I recommend this manuscript be published in the Journal of " Nanomaterials" with the following modifications.
- Specific quantitative results should be given in the abstract for the physical characterization and piezoelectric properties of the device.
Following the reviewer's recommendations, the most notable quantitative results from the electrical and physical characterization of the materials have been included in the abstract of the article.
- In the Introduction section the authors need to present the technical problems to be solved in comparison with previous work and the innovation points of this study.
Following the reviewer's recommendations, the introduction has been extensively expanded to justify the motivation and innovation of this study.
- The author should combine the information related to the materials and testing instruments in a separate paragraph.
Following the reviewer's instructions, the characterization section has been split into two separate sections: Materials and Methods, and Characterization.
- The labels in Figures 2a-b and 3b cover part of the curve and need to be modified.
Following the reviewer's instructions, the labels on the graphs of Figures 2a-b and b have been adjusted to ensure they do not cover any of the curves. Additionally, the colors of the graphs have been modified to enhance visibility.
- The scales of 6.4um and 8.5um in Figure 9a are the same length and need to be checked and corrected.
Following the reviewer's instructions, the scales of Figure 9a have been corrected.
- The tick marks for the upper and right axes should be removed.
Following the reviewer's instructions, the verification marks on the upper and right axes of figures 3a-b, 4a-b, 8b-c, 9b-d, 11c-d, and 12c have been removed.

Round 2
Reviewer 1 Report
Thank you
Enough
Reviewer 2 Report
Accept it in its current form
Accept it in its current form